# Let Occ Flow: Self-Supervised
# 3D Occupancy Flow Prediction

**Yili Liu**[1*]     **Linzhan Mou**[1*]     **Xuan Yu**[1]     **Chengrui Han**[1]
**Sitong Mao**[2]     **Rong Xiong**[1]     **Yue Wang**[1†]
[1]Zhejiang University     [2]Huawei Cloud Computing Technologies Co., Ltd.
{ylliu01, moulz}@zju.edu.cn

**Abstract:** Accurate perception of the dynamic environment is a fundamental task for autonomous driving and robot systems. This paper introduces Let Occ Flow, the first self-supervised work for joint 3D occupancy and occupancy flow prediction using only camera inputs, eliminating the need for 3D annotations. Utilizing TPV for unified scene representation and deformable attention layers for feature aggregation, our approach incorporates a novel attention-based temporal fusion module to capture dynamic object dependencies, followed by a 3D refine module for fine-gained volumetric representation. Besides, our method extends differentiable rendering to 3D volumetric flow fields, leveraging zero-shot 2D segmentation and optical flow cues for dynamic decomposition and motion optimization. Extensive experiments on nuScenes and KITTI datasets demonstrate the competitive performance of our approach over prior state-of-the-art methods. Our project page is available at https://eliliu2233.github.io/letoccflow/.

**Keywords:** 3D occupancy prediction, occupancy flow, Neural Radiance Field

## 1 Introduction

Accurate perception of the surrounding environment is crucial in autonomous driving and robotics for downstream planning and action decisions. Recently, vision-based 3D object detection [1, 2] and occupancy prediction [3, 4, 5, 6, 7] have garnered significant attention due to their cost-effectiveness and robust capabilities. Building on these foundations, the concept of occupancy flow prediction has emerged to offer a comprehensive understanding of the dynamic environment by estimating the scene geometry and object motion within a unified framework [8, 9].

Existing occupancy flow prediction methods [8, 9] rely heavily on detailed 3D occupancy flow annotations, which complicates their scalability to extensive training datasets. Previous research has attempted to address the dependency on 3D annotations for occupancy prediction tasks. Some strategies utilized LiDAR data to generate occupancy labels [3, 10, 11], but struggle to balance between accuracy and completeness due to LiDAR's inherent sparsity and dynamic object movements. Recently, some methods have emerged to enable 3D occupancy predictions in a self-supervised manner by integrating the NeRF-style differentiable rendering with either weak 2D depth cues [7] or photometric supervision [5, 6]. However, existing self-supervised techniques take static world assumption as a premise and never consider the motion of foreground objects. This raises a question: *Can we bridge 2D perception cues to enable self-supervised occupancy flow prediction?*

To address the aforementioned issues, we introduce Let Occ Flow, a novel method for simultaneously predicting 3D occupancy and occupancy flow using only camera input, without requiring any 3D annotations for supervision, as shown in Figure 1. Our approach employs Tri-perspective View(TPV) [11] as a unified scene representation and incorporates deformable attention layers to aggregate multi-view image features. For temporal sequence inputs, we utilize an attention-based

---

*Equal contribution. †Corresponding author.

8th Conference on Robot Learning (CoRL 2024), Munich, Germany.

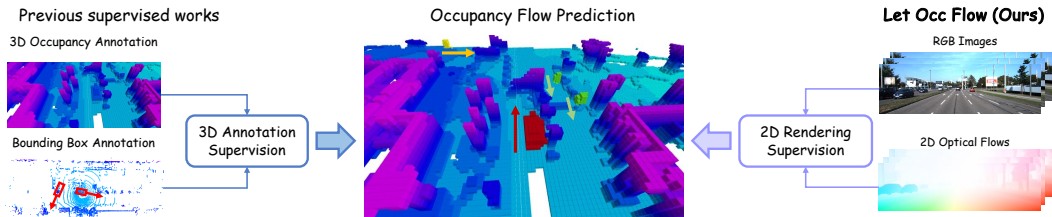

Figure 1: **Let Occ Flow** introduces a novel self-supervised training paradigm for 3D occupancy and occupancy flow prediction. Unlike earlier studies that relied on expensive annotations for 3D occupancy flow, our method employs readily available 2D labels to train the occupancy flow network.

temporal fusion module to effectively capture long-distance dependencies of dynamic objects. Subsequently, we use a 3D refine module for spatial volume feature aggregation and transform the scene representation into fine-grained volumetric fields of SDF and flow.

Differentiable rendering techniques are then applied to jointly optimize scene geometry and object motion. Specifically, for geometry optimization, we employ a photometric consistency supervision [6] and optional range rendering supervision [12, 13]. For motion optimization, we leverage semantic and optical flow cues for dynamic decomposition and supervision. These cues are convenient to obtain from pre-trained 2D models, which have shown superior zero-shot generalization capability in comparison to 3D models. Furthermore, previous studies have demonstrated that self-supervised photometric consistency [14, 15] enables the pre-trained flow estimator to effectively narrow the generalization gap across different data domains. Specifically, we employ a pre-trained 2D open-vocabulary segmentation model GroundedSAM [16] to derive a segmentation mask of movable objects. In addition, we use optical flow cues from an off-the-shelf 2D estimator [17, 18] to offer effective supervision of pixel correspondence for joint geometry and motion optimization.

In summary, our main contributions are as follows:

1. We proposed Let Occ Flow, the first self-supervised method for jointly predicting 3D occupancy and occupancy flow, by integrating 2D optical flow cues into geometry and motion optimization.

2. We designed a novel attention-based temporal fusion module for efficient temporal interaction. Furthermore, we proposed a flow-oriented optimization strategy to mitigate the training instability and sample imbalance problem.

3. We conducted extensive experiments on various datasets with qualitative and quantitative analyses to show the competitive performance of our approach.

## 2 Related Work

**3D Occupancy Prediction.** Vision-based 3D occupancy prediction has emerged as a promising method to perceive the surrounding environment. TPVFormer [11] introduces a tri-perspective view representation from multi-camera images. SurroundOcc [3] utilizes sparse LiDAR data to generate dense occupancy ground truth as supervision. Rather than using 3D labels, RenderOcc [7] employs volume rendering to enable occupancy prediction from weak depth labels. Recent works [4, 5, 6] propose a LiDAR-free self-supervised manner by photometric consistency. Despite the progress in static occupancy prediction, there has been comparatively less focus on occupancy flow prediction—a critical component for dynamic planning and navigation in autonomous driving.

**Neural Radiance Field.** Reconstructing 3D scenes from 2D images remains a significant challenge in computer vision. Among the various approaches, Neural Radiance Field (NeRF)[19] has gained increasing popularity due to its strong representational ability. To enhance training efficiency and improve rendering quality, several methods[20, 21, 22] have adopted explicit voxel-based representations. Additionally, some approaches [23, 24] introduced NeRF into surface reconstruction by developing Signed Distance Function(SDF) representation. Further advancements [25, 26, 27, 28]

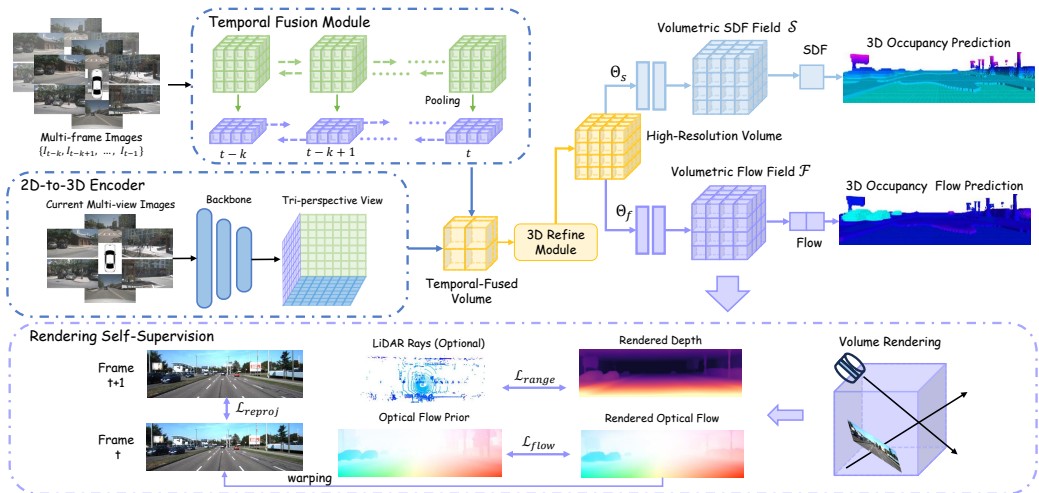

Figure 2: **The overall architecture of Let Occ Flow**. We employ deformable-attention layers to integrate multi-view image input into TPV representation. The temporal fusion module utilizes BEV-based backward-forward attention to fuse temporal feature volumes. The 3D Refine Module further aggregates spatial features and upsample the fused volume into a high-solution representation. Then we apply two separate MLP decoders to construct volumetric SDF and flow fields, and finally perform self-supervised occupancy flow learning utilizing reprojection consistency, optical flow cues, and optional LiDAR ray supervision via differentiable rendering.

have been made to facilitate scene-level sparse view reconstruction by conditioning NeRF on image features, laying the groundwork for generalizable NeRF architectures.

**Dynamic Driving Scene Reconstruction.** Recent approaches proposed to combine flow prediction with vision-based occupancy network [8, 9] for dynamic scene reconstruction, while they rely heavily on complete 3D occupancy and flow annotations. Previous approach [13] has successfully applied NeRF into self-supervised learning of dynamic driving scene reconstruction, while it typically requires expensive per-scene optimization. To the best of our knowledge, we are the very first work to achieve generalizable dynamic driving scene reconstruction in a self-supervised manner.

## 3 Method

The overall framework is shown in Figure 2. In Sec.3.2, we first extract 3D volume features $V$ from multi-view temporal sequences with 2D-to-3D feature encoder. To capture the geometry and motion information, we employ an attention-based temporal fusion module and a 3D refine module to ensure effective spatial-temporal interaction. In Sec.3.3, we construct the volumetric SDF and flow field using two MLP decoders and perform differentiable rendering to optimize the network in a self-supervised manner. Finally in Sec.3.4, we illustrate a flow-oriented two-stage training strategy and dynamic disentanglement scheme for joint geometry and motion optimization.

### 3.1 Problem Formulation

We aim to predict 3D occupancy $O_t$ and occupancy flow $F_t$ of the surrounding scenes at timestep $t$ using a temporal sequence of multi-view camera inputs. Formally, the task is represented as:

$$O_t, F_t = \mathcal{G}(I_{t-k}, I_{t-k+1}, ..., I_t) \tag{1}$$

where $\mathcal{G}$ is a neural network, $I_t$ represents $N$-view images $\{I_t^1, I_t^2, ..., I_t^N\}$ at timestamp $t$. $O_t \in \mathbb{R}^{H \times W \times Z}$ denotes the 3D occupancy at timestamp $t$, representing the occupied probability of each 3D volume grid with the value between 0 and 1. And $F_t \in \mathbb{R}^{H \times W \times Z \times 2}$ denotes the occupancy flow at timestamp $t$, representing the object motion vector excluding ego-motion of each grid in the horizontal direction of 3D space, as described in OccNet [8].

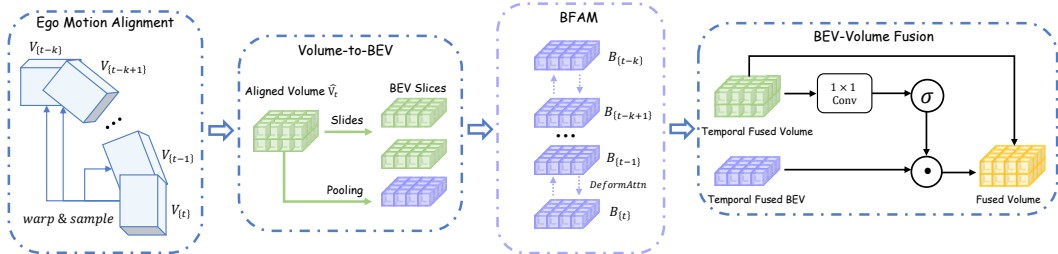

Figure 3: **Architecture of temporal fusion module**. The temporal fusion module consists of ego-motion alignment and temporal feature aggregation. While ego-motion alignment is employed in voxel space to align temporal feature volumes, BEV-based temporal feature aggregation leverages deformable attention with a backward-forward process to achieve temporal interaction.

### 3.2 2D-to-3D Encoder

**2D-to-3D Feature Encoder.** Following existing 3D occupancy methods [6, 11], we utilize TPV to effectively construct a unified representation from multi-view image input, which pre-define a set of queries in the three axis-aligned orthogonal planes:

$$T = \{T^{HW}, T^{ZH}, T^{WZ}\}, \quad T^{HW} \in \mathbb{R}^{H \times W \times C}, T^{ZH} \in \mathbb{R}^{Z \times H \times C}, T^{WZ} \in \mathbb{R}^{W \times Z \times C} \quad (2)$$

where $H, W, Z$ denote the resolution of the three planes and $C$ represents the feature dimension. We can integrate multi-view multi-scale image features $F$ extracted from the 2D encoder backbone [29] into triplane using deformable cross-attention [30] (DAttn) and feed forward network (FFN):

$$T^{p_i} = FFN(DAttn(DAttn(T^{p_i}, \{T^{p_j}\}_{p_j \in P - \{p_i\}}), F)), \quad p_i \in P = \{HW, ZH, WZ\} \quad (3)$$

Then we construct a full-scale 3D feature volume $V$ by broadcasting each TPV plane along the corresponding orthogonal direction and aggregating them by summation up [11].

**Temporal Fusion.** To effectively capture temporal geometry and object motion, we design a temporal fusion module to enhance the scene representation further. Specifically, our temporal fusion module integrates historical feature volume into the current frame. As depicted in Figure 3, this module comprises two main components: ego-motion alignment in voxel space and a Backward-Forward Attention Module (BFAM) in BEV space. To compensate for ego-motion, we first apply feature warping by transforming historical feature volumes to the current frame, according to the relative pose in ego coordinate:

$$\hat{V}_{t_i} = \langle V_{t_i}, T_t^{t_i} \cdot P_t \rangle, \quad t_i \in \{t - k, t - k + 1, ..., t - 1\} \quad (4)$$

where $\hat{V}_{t_i} \in \mathbb{R}^{H \times W \times Z \times C}$ is the aligned feature volume in the current local ego coordinate. $P_t \in \mathbb{R}^{H \times W \times Z \times 3}$ denotes the predefined reference points of current feature volume, $T_t^{t_i}$ is the ego pose transformation matrix from $t$ to $t_i$, and $\langle \cdot, \cdot \rangle$ indicates the linear interpolation operator.

Inspired by prior works [31, 32], we compress the aligned feature volume $\hat{V}_t$ into a series of BEV slices $B_t \in \mathbb{R}^{H \times W \times C}$ along the z-axis and employ deformable attention [30] (DAttn) between these BEV slices in the horizontal direction. To effectively capture overall information in the vertical dimension, we derive a global BEV feature $B_g$ using an average pooling operator and integrate it into the BEV slices. Furthermore, we employ a Backward-Forward Attention Module (BFAM) to enhance the interaction between temporal BEV features:

$$B_{t_j} = B_{t_j} + \beta \cdot DAttn(B_{t_j}, \{B_{t_i}, B_{t_j}\}) \quad (5)$$

where $B_{t_i}, B_{t_j}$ represents the BEV features at adjacent frames $t_i, t_j$, and $\beta$ is a learnable scale factor. In the backward process, $t_j = t_i - 1$, while in the forward process, $t_j = t_i + 1$.

After BFAM, we recover the fused feature volume $\tilde{V}$ by merging fused BEV slices, and utilize the BEV-Volume Fusion module to achieve interaction between fused global BEV features $\tilde{B}_g$ and fused feature volume $\tilde{V}$ for the final feature volume $\tilde{V}'$:

$$\tilde{V}' = \tilde{V} + \sigma(conv(\tilde{V})) \cdot unsqueeze(\tilde{B}_g, -1) \quad (6)$$

where $conv$ is a $1 \times 1$ convolution module and $\sigma$ is a sigmoid function.

**3D Refine Module.** Following the acquisition of the final fused feature volume $\tilde{V}'$, we utilize a residual 3D convolution network to further aggregate spatial features and subsequently upsample it to refined feature volume $V_{refine}$ with high resolution using 3D deconvolutions.

## 3.3 Rendering-Based Optimization

**Rendering Strategy.** Given refined feature volume $V_{refine}$, we employ two separate MLP $\{\Theta_s, \Theta_f\}$ to construct volumetric SDF field $\mathcal{S}$ and volumetric flow field $\mathcal{F}$:

$$\mathcal{S} = \Theta_s(V_{refine}), \quad \mathcal{F} = \Theta_f(V_{refine}) \tag{7}$$

During training, we employ a rendering-based strategy to concurrently optimize the 3D occupancy and occupancy flow without 3D annotations. Following [4, 6], we utilize SDF for geometric representation, with occlusion-aware rendering as NeuS[23]. Details are shown in the supplementary.

**Rendering-Based Supervision.** Inspired by [4, 5, 6, 25], we utilize a reprojection photometric loss for effective optimization of the volumetric SDF field:

$$\mathcal{L}_{reproj}(x) = \sum_{i=1}^{N} w_i \mathcal{L}_{photo}(\langle I_t, x \rangle, \langle I_{t+1}, KT^{-1}T_t^{t+1}(TK^{-1}[x, d_i] + f_i) \rangle) \tag{8}$$

where $N$ denotes the number of points sampled along the ray, $x$ is the sampled pixel on the current image, $T_t^{t+1}$ represents the transformation matrix from timestamp $t$ to $t + 1$ in ego coordinate, $K$ and $T$ are the intrinsic and extrinsic parameters of cameras, respectively. Additionally, $w_i$, $d_i$, and $f_i$ denote the weight, depth, and flow of $i^{th}$ sampled point. For the photometric loss $\mathcal{L}_{photo}$, we utilize the combination of L1 and D-SSIM loss, following monodepth [33].

Though photometric loss is sufficient for geometric supervision, optimizing object motion jointly remains a significant challenge. Therefore, we incorporate optical flow cues from an off-the-shelf flow estimator [17, 18] to offer an additional source of supervision for geometry and motion estimation.

$$\mathcal{L}_{flow}(x) = \sum_{i=1}^{N} w_i \| \langle O_{t \to t+1}, x \rangle, KT^{-1}T_t^{t+1}(TK^{-1}[x, d_i] + f_i) - x \|_1 \tag{9}$$

where $O_{t \to t+1}$ represents the estimated forward flow map from frame $t$ to $t + 1$.

When LiDAR is available, we could combine LiDAR supervision to enhance the geometry. Specifically, we first independently generate rays from the LiDAR measurement to ensure an effective sampling rate in regions with sufficient geometric observations, thereby avoiding occlusion problems from camera extrinsic. Furthermore, we implement a Scale-Invariant loss $\mathcal{L}_{range}$ [34] between rendered ranges and LiDAR measurements to mitigate the global scene scale influence.

## 3.4 Flow-Oriented Optimization

Due to the training instability caused by the inherent ambiguities in jointly optimizing geometry and motion without ground truth labels, and the sample imbalance problem of moving objects and the static background, it becomes imperative to employ an effective flow-oriented optimization strategy.

**Two Stage Optimization.** To tackle the training instability challenge, we introduce a two-stage optimization strategy. Initially, we optimize the SDF field with the static assumption following the method proposed by SelfOcc [6] to generate a coarse geometric prediction. In the second stage, we jointly optimize the volumetric SDF and flow field with extra 2D optical flow supervision.

**Dynamic Disentanglement.** To effectively regularize the motion in static regions and solve the sample imbalance problem, we propose to conduct dynamic disentanglement using dynamic optical flow $f_x^d$, which is subtracted by flow cues $f_x^{opt}$ and static flow $f_x^s$ computed by rendered depth $d$:

$$f_x^d = f_x^{opt} - f_x^s = \langle O_{t \to t+1}, x \rangle - (KT^{-1}T_t^{t+1}TK^{-1}[x, d] - x) \tag{10}$$

To mitigate incorrect decomposition caused by inaccurate depth estimation in background regions (eg. sky and thin structure), we leverage the zero-shot open-vocabulary 2D semantic segmentation model, Grounded-SAM [16], to extract binary masks for movable foreground objects. With flow error mask thresholded by dynamic flow $f_x^d$, we could get an accurate dynamic mask for scene decomposition. Then we can regularize the volumetric flow field by minimizing the rendered flow map in static regions: $\mathcal{L}_{dreg} = \| \sum_m^{M_s} \sum_i^N w_i f_i \|_1$.

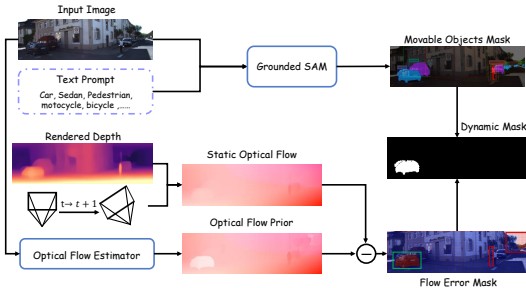

Figure 4: Our effective dynamic disentanglement scheme provides accurate dynamic mask for joint geometry and motion optimization.

Compared to the static background, dynamic objects such as cars or pedestrians account for a small proportion in driving scenes but hold significant importance for driving perception and planning. To address the imbalance caused by moving objects, we employ a scale factor to adjust the flow loss of dynamic rays based on their occurrence.

$$\mathcal{L}_{flow} = \mathcal{L}_{flow}^s + \gamma \mathcal{L}_{flow}^d, \quad \gamma = min(M_{all}/M_d, thre) \tag{11}$$

where $\mathcal{L}_{flow}^s$ and $\mathcal{L}_{flow}^d$ denote the static and dynamic parts of the flow loss $\mathcal{L}_{flow}$ calculated by Equation 9, while $M_{all}$ and $M_d$ represent the number of all sampled rays and dynamic rays, and $thre$ is a threshold hyperparameter.

**Static Flow Supervision.** Since optical flow cues provide the pixel correspondence between adjacent frames, they could be used to supervise the scene geometry effectively. Specifically, we utilize the dynamic mask mentioned above to decouple the static regions and then leverage the forward static flow for geometry supervision. To effectively use the abundant training data in self-supervised training, we sample an auxiliary previous frame based on the Gaussian distribution of the distance between the ego coordinates to offer an additional backward flow for geometric supervision.

Finally, with the SDF regularization loss $L_{eik}, L_H, L_{smooth}$, our overall loss function is as follows:

$$\mathcal{L} = \mathcal{L}_{reproj} + \lambda_{flow}\mathcal{L}_{flow} + \lambda_{range}\mathcal{L}_{range} + \lambda_{dreg}\mathcal{L}_{dreg} + \lambda_{eik}\mathcal{L}_{eik} + \lambda_H\mathcal{L}_H + \lambda_{smooth}\mathcal{L}_{smooth} \tag{12}$$

## 4 Experimental Results

### 4.1 Self-supervised 3D Occupancy Prediction

We conduct experiments on 3D occupancy prediction using the SemanticKITTI [35] dataset without LiDAR supervision and employ RayIoU [36] as evaluation metrics. We also evaluate the results of monocular depth estimation using Abs Rel, Sq Rel, RMSE, RMSE log as metrics.

As reported in Table 1, our method sets the new state-of-the-art for 3D occupancy prediction and depth estimation tasks without any form of 3D supervision compared with other supervised and self-supervised approaches. Thanks to our effective spatial-temporal feature aggregation and the integration of optical flow cues for supervision, our method greatly enhances the geometric representation capabilities, compared to other rendering-based methods [5, 6, 25].

Table 1: **3D occupancy prediction and depth estimation on SemanticKITTI [35] dataset.** MonoScene [38] is trained with 3D supervision and cannot render monocular depth. We adapt the OccNeRF [5] to the SemanticKITTI [35] dataset based on the official implementation.

| Method | Occupancy | | | | Depth | | | |
|---|---|---|---|---|---|---|---|---|
| | RayIoU$_{1m, 2m, 4m}$↑ | | | RayIoU↑ | Abs Rel↓ | Sq Rel↓ | RMSE↓ | RMSE log↓ |
| MonoScene [38] | 26.00 | 36.50 | 49.09 | 37.19 | - | - | - | - |
| SceneRF [25] | 20.65 | 35.74 | 56.35 | 37.58 | 0.146 | 1.048 | 5.240 | 0.241 |
| OccNeRF [5] | 23.22 | 40.17 | 62.57 | 41.99 | 0.132 | 1.011 | 5.497 | 0.233 |
| SelfOcc [6] | 23.13 | 35.38 | 50.31 | 36.27 | 0.152 | 1.306 | 5.386 | 0.245 |
| **Ours** | **28.62** | **45.60** | **66.95** | **47.06** | **0.117** | **0.817** | **4.816** | **0.215** |

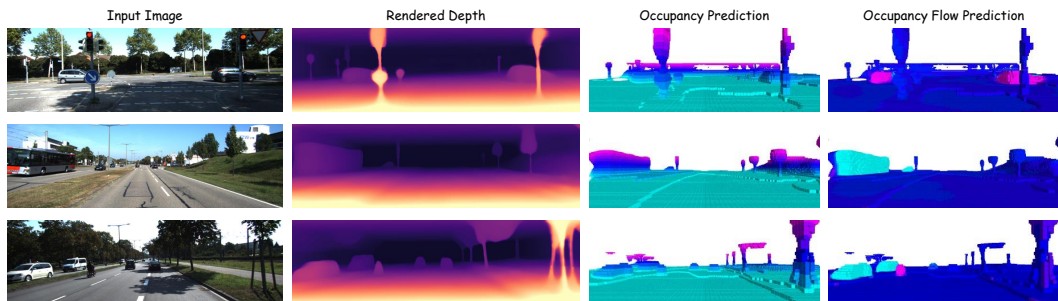

| Input Image | Rendered Depth | Occupancy Prediction | Occupancy Flow Prediction |

Figure 5: **Visualization results for depth estimation, 3D occupancy and occupancy flow prediction on the KITTI [37] dataset.** Our method can predict visually appealing depth maps, fine-grained occupancy, and accurate dynamic decomposition and motion estimation.

## 4.2 Self-supervised Occupancy and Occupancy Flow Prediction

We conduct experiments on 3D occupancy and occupancy flow prediction on the KITTI-MOT [37] and nuScenes [39] datasets. We use OccNeRF [5] and RenderOcc [7] as rendering-based baselines. To compare with OccNeRF, we employ a concatenation&conv temporal fusion module as in RenderOcc. We also add a flow head to these two methods with extra 2D optical flow labels as supervision for fair comparison. To compare with OccNet [8], we use their official implementation to reproduce the results with GT 3D occupancy flow labels from OpenOcc [8] as supervision.

Due to the absence of occupancy flow annotations, we report the occupancy flow evaluation metrics on the KITTI-MOT dataset [37] using LiDAR-projected depths and generated flow labels. More details are presented in the supplementary materials. On the nuScenes [39] dataset, with annotated occupancy flow ground truth, we report the RayIoU [36] and mAVE metrics as in OccNet [8].

Table 2: **Self-supervised occupancy and occupancy flow prediction on KITTI-MOT [37].**

| Method | Supervision | 3D Occupancy & Occupancy Flow | | | | | | |
|---|---|---|---|---|---|---|---|---|
| | | DE↓ | EPE↓ | DE_FG↓ | EPE_FG↓ | D_5%↓ | Fl_10%↓ | SF_10%↓ |
| OccNeRF-C* [5] | C | 3.547 | 6.986 | 7.976 | 10.445 | 0.204 | 0.272 | 0.306 |
| **Ours-C** | C | 3.431 | 3.529 | 6.429 | 7.635 | 0.215 | 0.084 | 0.118 |
| OccNeRF-L* [5] | C+L | 2.667 | 8.131 | 4.040 | 11.284 | **0.137** | 0.309 | 0.326 |
| **Ours-L** | C+L | **2.610** | **3.488** | **3.106** | **7.510** | 0.154 | **0.083** | **0.105** |

Table 3: **3D occupancy and occupancy flow prediction on nuScenes [39].**

| Method | Supervision | 3D Occupancy & Occupancy Flow | | | | |
|---|---|---|---|---|---|---|
| | | RayIoU_{1m, 2m, 4m}↑ | | | RayIoU↑ | mAVE↓ |
| OccNet [8] | 3D | **29.28** | **39.68** | 50.02 | 39.66 | 1.61 |
| OccNeRF-C* [5] | C | 9.93 | 19.06 | 35.84 | 21.61 | 1.53 |
| **Ours-C** | C | 17.49 | 28.52 | 44.33 | 30.12 | **1.42** |
| RenderOcc* [7] | L | 20.27 | 32.68 | 49.92 | 36.67 | 1.63 |
| OccNeRF-L* [5] | C+L | 16.62 | 29.25 | 49.17 | 31.68 | 1.59 |
| **Ours-L** | C+L | 25.49 | 39.66 | **56.30** | **40.48** | 1.45 |

We report the results in Table 2 and Table 3, C and L denote Camera and Lidar supervision. Compared to rendering-based methods [5, 7], our approach performs much better on both KITTI-MOT [37] and nuScenes [39], owing to our effective temporal fusion module and flow-oriented optimization strategy. Compared to 3D supervised OccNet [8], our approach achieves comparable performance on nuScenes [39], validating the effectiveness of our self-supervised training paradigm.

## 4.3 Ablation Study

We systematically conduct comprehensive ablation analyses on the temporal fusion module, optimization strategy, and static flow supervision to validate the effectiveness of every component of our method. All experiments are trained for 16 epochs, and without LiDAR supervision by default.

Table 4: **Temporal fusion module ablation on KITTI-MOT [37] dataset.**

| Temporal Fusion Module | DE↓ | EPE↓ | DE_FG↓ | EPE_FG↓ | D_5%↓ | Fl_10%↓ | SF_10%↓ | Params.↓ |
|---|---|---|---|---|---|---|---|---|
| Concatenation&3D Conv. | 3.515 | 3.638 | 6.423 | 7.743 | 0.223 | 0.090 | 0.126 | 11.39M |
| BEV Pooling & Forward Attn. | 3.600 | 3.763 | **6.339** | 7.971 | 0.228 | 0.091 | 0.124 | 5.21M |
| BEV Pooling & BFAM | 3.462 | 3.583 | 6.409 | 7.673 | **0.211** | 0.089 | 0.122 | 6.62M |
| Ours | **3.431** | **3.529** | 6.429 | **7.635** | 0.215 | **0.084** | **0.118** | 6.62M |

Table 5: **Optimization strategy ablation on KITTI-MOT [37] dataset.** Since the absence of two-stage optimization brings instability during training, we use extra LiDAR ray supervision here.

| Two Stage Optimization | Dynamic Disentanglement | DE↓ | EPE↓ | DE_FG↓ | EPE_FG↓ | D_5%↓ | Fl_10%↓ | SF_10%↓ |
|---|---|---|---|---|---|---|---|---|
| | | 3.760 | 4.403 | 4.194 | 8.853 | 0.258 | 0.121 | 0.152 |
| ✓ | | 3.064 | 3.633 | 3.600 | 7.966 | 0.191 | 0.089 | 0.114 |
| ✓ | ✓ | **2.610** | **3.488** | **3.106** | **7.510** | **0.154** | **0.083** | **0.105** |

Table 6: **Static flow supervision ablation on KITTI-MOT [37] dataset.**

| Backward Static Flow Supervision | Forward Static Flow Supervision | DE↓ | EPE↓ | DE_FG↓ | EPE_FG↓ | D_5%↓ | Fl_10%↓ | SF_10%↓ |
|---|---|---|---|---|---|---|---|---|
| | | 6.715 | 4.128 | 6.598 | 8.160 | 0.239 | 0.099 | 0.149 |
| | ✓ | 3.495 | 3.614 | 6.510 | 7.987 | 0.217 | **0.084** | 0.123 |
| ✓ | ✓ | **3.431** | **3.529** | **6.429** | **7.635** | 0.215 | 0.084 | **0.118** |

**Temporal Fusion.** We report the results of different temporal fusion settings in Table 4. Compared to the 3D Convolution module in [7], our method achieves better performance with lower memory cost due to our efficient BEV-based attention mechanism and the effective long-distance temporal interaction. Moreover, our BFAM module outperforms the forward-only attention method introduced in [8] due to its enhanced temporal interaction capabilities. Additionally, Our method effectively incorporates local and global information in the height dimension compared to BEV-only attention.

**Optimization Strategy.** We investigate the effect of our optimization strategy in Table 5. It is shown that both two-stage optimization and dynamic disentanglement scheme benefit mitigating the impact of the ambiguities for joint optimization of scene geometry and object motion.

**Static Flow Supervision.** We verify the effectiveness of static flow supervision in Table 6. As reported, extra pixel correspondence priors from forward and backward static flow can effectively improve the quality of scene geometry, thus making occupancy flow better.

## 5   Discussion

**Limitation and Future work.** Although we use temporal sequence input to better exploit the historical information, our model cannot completely handle the occlusion problem due to the inherent rendering-based limitation. Subsequent research could investigate long-term occupancy flow modeling and solutions to leverage the temporal sequence supervision to scale up the visible range of perspective. In addition, the accuracy of occupancy flow prediction relies on the quality of optical flow cues. Future work can pay more attention on the improvement of the flow supervision quality. Finally, our occupancy flow prediction does not explicitly enforce consistency within instances, and future work may explore to integrate instance perception into occupancy flow prediction.

**Conclusion.** We introduced Let Occ Flow, the first self-supervised method for vision-based 3D occupancy and occupancy flow prediction. Utilizing the effective temporal fusion module and flow-oriented optimization, our approach effectively captures dynamic object dependencies, thus enhancing both scene geometry and object motion. Furthermore, our method extends the differentiable rendering to the volumetric flow field for self-supervised learning. Consequently, our work paves the way for future research into more efficient and accurate self-supervised learning frameworks in dynamic environment perception.

**Acknowledgments**

We sincerely thank the anonymous reviewers for their helpful comments in revising the paper. This work was supported in part by the National Nature Science Foundation of China under Grant 62373322, and in part by Zhejiang Provincial Natural Science Foundation of China under Grant No. LD24F030001.

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

# Supplementary Material

Our supplementary material includes dataset details, implementation details, evaluation metrics, and additional visualization results.

## A  Dataset Details

**The SemanticKITTI [35] dataset** is a large-scale dataset based on the KITTI Vision Benchmark [37], which includes automotive LiDAR scans voxelized into $256 \times 256 \times 32$ grids with 0.2m resolution for 22 sequences. The dataset contains 28 classes with non-moving and moving objects. In our experiments, we only use RGB images captured by cam2 as input from the KITTI odometry benchmark and follow the official split of the dataset, with sequence 08 for validation and other sequences from 0 to 10 for training.

**The KITTI-MOT [37] dataset** is based on the KITTI Multi-Object Tracking Evaluation 2012, which consists of 21 training sequences and 29 test sequences. It holds stereo images from two forward-facing cameras and LiDAR point clouds. We selected all the training sequences except sequences where the ego-car is stationary or contains only a small number of moving objects(sequence 0012, 0013, and 0017) for training, and select testing sequences 0000-0014 for validation. We also apply LiDAR data and pseudo-optical flow to coarsely select dynamic frames for a higher sampling rate in our training process.

**The nuScenes [39] dataset** consists of 1000 sequences of various driving scenes of 20 seconds duration each and the keyframes are annotated at 2Hz. Each sample includes RGB images collected by 6 surround cameras with $360°$ horizontal FOV and LiDAR point clouds from 32-beam LiDAR sensors. The 1000 scenes are officially divided into training, validation, and test splits with 700, 150, and 150 scenes, respectively.

## B  Implementation Details

### B.1  Model Architecture

We adopted ConvNeXt-Base with SparK [40] pretraining as the 2D image backbone and a four-level FPN [41] to extract image features. We utilize TPV [11] uniformly divided to represent a cuboid area for multi-view feature integration, i.e. [80, 80, 6.4] meters around the ego car for nuScenes [39] and [51.2, 51.2, 6.4] meters in front of the ego car for SemanticKITTI [35] and KITTI-MOT [37]. Considering the computational cost of the subsequent temporal fusion module, we use half of the output occupancy resolution for TPV grid cell, i.e. 0.8 meters for nuScenes and 0.4 meters for SemanticKITTI and KITTI-MOT, respectively.

For the input temporal sequence, we utilize 2 previous frames for our temporal fusion module. Our 3D refine module includes a three-layer residual 3D convolution block and a 3D FPN block for spatial feature integration. To upsample the volume into the output occupancy resolution, we use a deconvolution module as FBOCC [42]. We adopt separate two-layer MLPs $\{\Theta_s, \Theta_f\}$ as decoders to construct volumetric fields for SDF and flow.

### B.2  2D Flow Estimation

For occupancy flow prediction in KITTI-MOT [37] dataset, we leverage the flow estimation model of Unimatch [17] trained on FlyingChairs [43], FlyingThings3D [44] and fine-tuned on KITTI [37] to predict optical flow maps for supervision directly. Note that the Unimatch model is trained in a supervised manner, we provide an unsupervised flow fine-tuning strategy following [45] when adapted to different driving scenes. Specifically, we fine-tune the Unimatch model utilizing unsupervised flow techniques including stopping the gradient at the occlusion mask, encouraging smoothness before upsampling the flow field, and continual self-supervision with image resizing.

As for the nuScenes [39] dataset, the 3D occupancy flow ground-truth data is only available at 2Hz keyframes so it is challenging for optical flow estimation. Thus we employed a tracking-based flow estimation strategy using CoTrackerV2 [18]. We aim to obtain accurate optical flow between two keyframes by including the non-keyframe sequences to capture the long-term track. Specifically, we first use open-vocabulary 2D segmentor GroundedSAM[16] to predict dynamic foreground semantic segmentation. Then we take the adjacent keyframes and the non-keyframe sequences between the two keyframes as video input and conduct dense track to capture the long-term motion dependency in the masked regions. Once we have the initial pixel coordinates in the first keyframe and the corresponding pixel coordinates in the next, we subtract the two and get the final optical flow map in the masked regions. We take this optical flow map with the dynamic foreground mask as pseudo-flow labels for occupancy flow prediction.

### B.3 Training Settings

The resolution of the input image is 512x1408 for nuScenes [39], 352x1216 for SemanticKITTI [35], and 352x1216 for KITTI-MOT [37]. For loss weight, we set $\lambda_{flow} = 5 \times 10^{-3}, \lambda_{eik} = 0.1, \lambda_{dreg} = 0.1$, if present, and the weights for the edge $\lambda_{edge}$ and the LiDAR $\lambda_{lidar}$ losses are 0.02 and 0.2 respectively. During training, we adopt the AdamW optimizer with an initial learning rate 1e-4 and weight decay of 0.01. We use the multi-step learning rate scheduler with linear warming up in the first 1k iterations. We train our models with a total batch size of 8 on 8 NVIDIA A100 80GB GPUs with 8 epochs for the first stage, and 16 epochs for the second stage. Experiments on SemanticKITTI [35] and KITTI-MOT [37] take less than one day, while experiments on nuScenes [39] finish within two days.

### B.4 Temporal Fusion Module

Algorithm 1 provides the pseudo-code of our proposed temporal fusion module. In the BEV fusion process, we employ a Backward Forward Attention Module (BFAM) with deformable attention (DAttn) to fuse the BEV features from two adjacent frames. To illustrate our algorithmic process clearly, we visually demonstrate our temporal fusion module in the demo video.

---

**Algorithm 1** Pseudo-code for Temporal Fusion Module

---

1: **Input**: A temporal sequence of Voxel features $\{V_{-(n-1)}, \ldots, V_{-1}, V_0\}$. $V_0$ represents Voxel feature of the current frame and $V_{-i}$ corresponds to the $i$-th frame before $V_0$. A sequence of transformation matrix of ego coordinates $\{T_{-(n-1)}, \ldots, T_{-1}, T_0\}$.
2: **for** $i = 0$ **to** $n - 1$ **do**
3:     $\hat{V}_{-i} \leftarrow \text{EgoMotionAlignment}(V_{-i}, T_{-i}, T_0)$
4:     $B^g_{-i} \leftarrow \text{MeanPooling}(\hat{V}_{-i})$
5:     $B_{-i} \leftarrow \text{Concatenate}(\text{VolumeToSlices}(\hat{V}_{-i}), B^g_{-i})$          ▷ temporal volumes to BEV slices
6: **end for**
7: **for** $i = 0$ **to** $n - 2$ **do**
8:     $B_{-(i+1)} \leftarrow B_{-(i+1)} + \beta \cdot \text{DAttn}(B_{-(i+1)}, \{B_{-i}, B_{-(i+1)}\})$          ▷ backward process
9: **end for**
10: **for** $i = n - 2$ **to** $0$ **do**
11:     $B_{-i} \leftarrow B_{-i} + \beta \cdot \text{DAttn}(B_{-i}, \{B_{-(i+1)}, B_{-i}\})$          ▷ forward process
12: **end for**
13: $\tilde{V}, \tilde{B}_g \leftarrow \text{SlicesToVolume}(B_0)$
14: $\tilde{V}' \leftarrow \text{BEVVolumeFusion}(\tilde{V}, \tilde{B}_g)$
15: **return** $\tilde{V}'$

---

### B.5 Differentiable Rendering

In the rendering stage, we conduct a uniform sampling of $N$ points $P = \{p_i | i = 1, ..., N\}$ along the ray and apply a tri-linear interpolation operation to efficiently compute the SDF values for each point

from the volumetric SDF field [22]. Furthermore, the unbiased rendering weights can be calculated by $w_i = \alpha_i \prod_{j=1}^{i-1}(1 - \alpha_j)$, with $\alpha_i$ denoting the opacity value proposed by NeuS [23]:

$$\alpha_i = \max\left(\frac{\Phi(s_i) - \Phi(s_{i+1})}{\Phi(s_i)}, 0\right) \tag{13}$$

where $\Phi(x)$ is sigmoid function $\Phi(x) = (1 + e^{-\xi x})^{-1}$ with a temperature coefficient $\xi$.

Let $d_i$ and $f_i$ denote the depth and flow of the i-th point, we can calculate the rendered depth $d$ and flow $f$ of the ray by:

$$d = \sum_{i=1}^{N} w_i d_i, \quad f = \sum_{i=1}^{N} w_i f_i \tag{14}$$

### B.6 Regularization.

Due to the explicit physical interpretation of SDF representation, regularization can be effectively applied to the volumetric SDF field to prevent over-fitting with limited supervision. Initially, an eikonal term is employed for each sample point along camera and LiDAR rays to promote the establishment of a valid SDF field:

$$\mathcal{L}_{eik} = (1 - \| \langle \nabla \mathcal{S}, \boldsymbol{p} \rangle) \|_2)^2 \tag{15}$$

where $\nabla \mathcal{S}$ denotes the pre-computed gradient field of the volumetric SDF grid. Additionally, we also utilize a Hessian loss $\mathcal{L}_H$ on the second-order derivatives of the SDF grid, following the approach of SelfOcc [6]. Furthermore, an edge-aware smoothness loss $\mathcal{L}_{smooth}$ is employed following monodepth2 [33] to enforce neighborhood consistency of rendered depths and flows.

## C Evaluation Metrics

### C.1 Depth Estimation Evaluation Metrics

Following [46, 47, 48], we use evaluation metrics for self-supervised depth estimation as follows:

$$\text{Abs Rel: } \frac{1}{|T|} \sum_{d \in T} |d - d^*| / d^*, \quad \text{Sq Rel: } \frac{1}{|T|} \sum_{d \in T} |d - d^*|^2 / d^* \tag{16}$$

$$\text{RMSE: } \sqrt{\frac{1}{|T|} \sum_{d \in T} |d - d^*|^2}, \quad \text{RMSE log: } \sqrt{\frac{1}{|T|} \sum_{d \in T} |\log d - \log d^*|^2} \tag{17}$$

where $d$ and $d^*$ indicate predicted and ground truth depths respectively, and $T$ indicates all pixels on the depth map. We calculate metrics for depth values in the range of [0.1, 80] meters using 1:2 resolution against the raw image.

### C.2 3D Occupancy Prediction Evaluation Metrics

Previous approaches [6, 25, 38] utilize the intersection over union (IoU) as the evaluation metrics of 3D occupancy prediction on SemanticKITTI [35] dataset. However, according to our experiment results, the penalty is overly strict in evaluating the reconstruction quality effectively. As illustrated in Figure 6, we observed that rendering-based methods [6, 25] often generate true positive (TP) predictions(marked in green in the figure) distributed on the ground or in invisible regions below the ground. This prevents effective evaluation of reconstruction details, as a deviation of one voxel will lead to an IoU of zero. Furthermore, rendering-based methods tend to predict a thick surface

due to the absence of supervision in invisible regions, causing a large number of false positive (FP) predictions(marked in red in the figure) and a significant reduction of the precision metric.

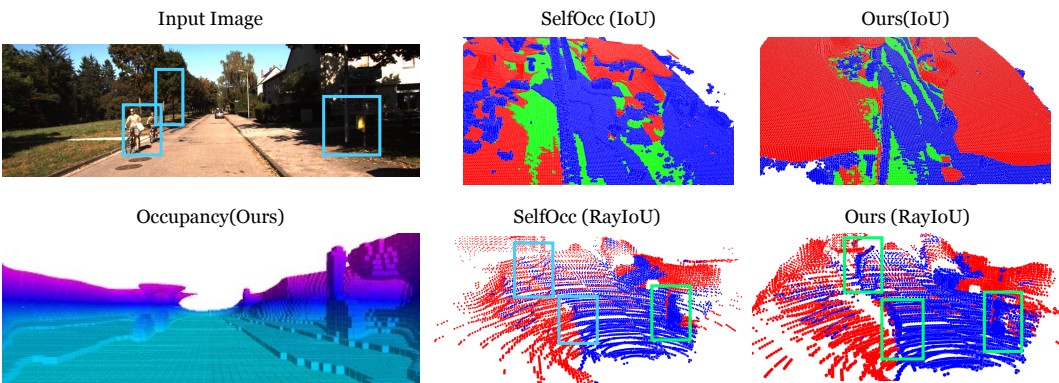

Figure 6: The comparison of IoU and RayIoU measurement results in 3D occupancy prediction.

To demonstrate the advanced occupancy prediction quality of our approach, we employ Ray-based IoU (**RayIoU**) [36] as our evaluation metric and re-evaluate other comparative methods using their open-source checkpoint models. Initially, we generated LiDAR rays with origins sampled from the sequence, and then compute the travel distances of each LiDAR ray before intersecting with the occupied voxels. A query ray is classified as true positive(TP) when the L1 error between the predicted depth and ground-truth depth is less than a pre-defined threshold (1m, 2m, and 4m). We finally average over distance thresholds and compute the mean RayIoU metric.

As is shown in Figure 6, the RayIoU metric reflected the superiority of our method on the detailed prediction of the tree and bicycles compared to SelfOcc.

### C.3 Occupancy Flow Prediction Evaluation Metrics

On the KITTI-MOT [37] data, due to the absence of ground-truth occupancy flow labels, we conduct the evaluation following the scene flow benchmark of KITTI dataset [37]. Specifically, we use LiDAR-projected depth and pseudo 2D optical flow cues to generate disparity and optical flow labels. And we computed end-point error of rendered and ground-truth disparity (DE) and optical flow (EPE). D_5% represent the percentage of disparity outliers with end-point error smaller than 4 pixels or 5% of the ground-truth disparity. Fl_10% is the percentage of optical flow outliers with end-point error smaller than 8 pixels or 10% of the ground-truth flow labels. And SF_10% is the percentage of scene flow outliers (outliers in either D_10% or Fl_10%). To better evaluate the scene flow in foreground regions, we also reported the foreground disparity error (DE_FG) and optical flow error (EPE_FG) by leveraging the semantic mask obtained from GroundedSAM [16].

On the nuScenes [39] dataset, since we have the ground-truth occupancy flow labels, we use Ray-based IoU (**RayIoU**), and absolute velocity error (**mAVE**) for occupancy flow, following the evaluation metrics of *Occupancy and Flow track for CVPR 2024 Autonomous Grand Challenge*.

For **RayIoU** measurement, we perform evaluation on the OpenOcc benchmark [8] as introduced in subsection C.2. We use **mAVE** to indicate the velocity errors for a set of true positives (TP) with a threshold of 2m distance. The absolute velocity error (AVE) is defined for 8 classes ('car', 'truck', 'trailer', 'bus', 'construction_vehicle', 'bicycle', 'motorcycle', 'pedestrian') in m/s.

## D  Additional Ablative Studies

### D.1  Flow Supervision

To demonstrate the effectiveness to introduce 2D optical flow supervision into our approach, we conducted a supplementary ablation by dropping the flow supervision.

Dropping flow supervision would result in zero flow predictions in dynamic regions. The ambiguity between camera and object motion makes the photometric loss insufficient for jointly predicting accurate geometry and motion. This challenge is even more pronounced in occupancy-level geometry and motion prediction.

As shown in Table 7, while depth-related metrics (DE, DE_FG, D_5% ) exhibit minimal variance, there is a notable increase in the End-Point Error of foreground objects (EPE_FG), the optical flow outlier rate(Fl_10%) and the scene flow outlier rate(SF_10%), underscoring the importance of flow supervision in our model.

Table 7: **Static flow supervision ablation on KITTI-MOT [37] dataset.**

| Optical Flow Supervision | DE↓ | EPE↓ | DE_FG↓ | EPE_FG↓ | D_5%↓ | Fl_10%↓ | SF_10%↓ |
|---|---|---|---|---|---|---|---|
| | **2.447** | 3.649 | **2.936** | 9.663 | **0.149** | 0.114 | 0.139 |
| ✓ | 2.610 | **3.488** | 3.109 | **7.510** | 0.154 | **0.083** | **0.105** |

## D.2 Optimization Strategy on nuScenes

To demonstrate the results of our ablations on different datasets, we conducted a supplementary ablation by employing nuScenes with $256 \times 704$ image resolution and a half-epoch setting (8 epochs). This additional analysis serves to underscore the effectiveness of our method.

As demonstrated in Table 8, the ablation results on nuScenes align well with those on KITTI-MOT. The joint optimization of geometry and motion without two-stage optimization results in poor geometric performance due to the inherent ambiguities of ego and object motion. The average velocity error (mAVE) only accounts for true positive predictions, thus underestimating our full model by disregarding incorrect occupancy predictions in dynamic regions. Additionally, Our proposed dynamic disentanglement strategy enhances the performance by introducing geometric regularization of static regions and addresses the imbalance problem.

Table 8: **Optimization strategy ablation on nuScenes [39] dataset.** Since the absence of two-stage optimization brings instability during training, we use extra LiDAR ray supervision here.

| Two Stage Optimization | Dynamic Disentanglement | 3D Occupancy & Occupancy Flow | | | | |
|---|---|---|---|---|---|---|
| | | RayIoU$_{1m, 2m, 4m}$↑ | | | RayIoU↑ | mAVE↓ |
| | | 10.57 | 21.05 | 41.15 | 24.26 | **1.36** |
| ✓ | | 16.44 | 30.53 | 50.51 | 32.49 | 1.59 |
| ✓ | ✓ | **22.95** | **36.82** | **54.00** | **37.92** | 1.56 |

# E   Visualization Results

## E.1 Depth Estimation

Figure 7 shows the qualitative comparison of depth estimation on the SemanticKITTI [35] validation set. The visualization results illustrate the successful prediction of detailed and accurate depth by our method compared to other rendering-based approaches due to our effective temporal fusion module and optical flow supervision.

## E.2 3D Occupancy Prediction

Figure 8 shows the qualitative comparison of 3D occupancy prediction on the SemanticKITTI [35] validation set. 3D occupancy are color-coded according to the height of the voxels. Our method achieves precise occupancy prediction for thin structures such as poles, trees, and cyclists. Also, it provides smooth predictions for cars and road surfaces compared to other supervised [38] and self-supervised [5, 6, 25] methods.

We provided more visualization results of depth estimation, novel view depth synthesis, and 3D occupancy prediction to illustrate the superiority of our method. As is shown in Figure 9, our method exhibited strong performance across these three tasks, which is due to the accurate prediction of the visible surfaces. In addition, our method with effective temporal fusion can provide occlusion perception beyond the visible surface, thus generating high-quality novel-view depth estimations and global-view occupancy predictions.

## E.3 Occupancy Flow Prediction

Figure 10 and Figure 11 shows the visualization results of the depth estimation and occupancy flow prediction tasks on the KITTI-MOT [37] and nuScenes [39] validation set. For flow visualization, the color and brightness represent the motion direction and magnitude respectively. Our proposed method can simultaneously provide accurate 3D occupancy and occupancy flow prediction.

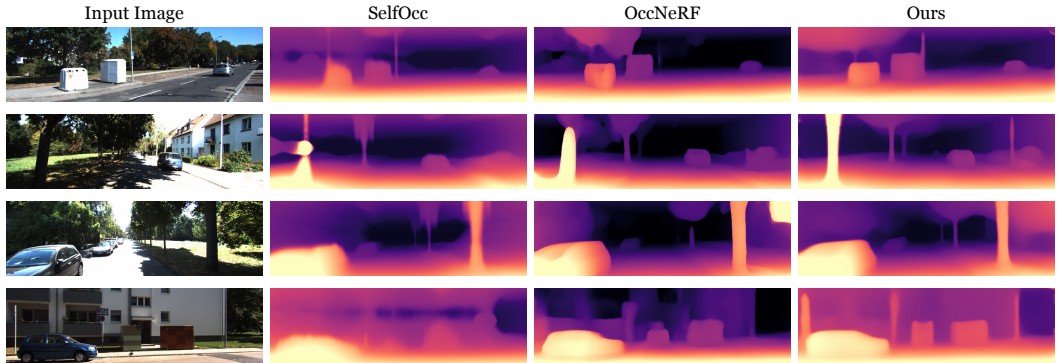

Figure 7: **Qualitative comparison for self-supervised depth estimation with other baselines on the SemanticKITTI [35] validation set.**

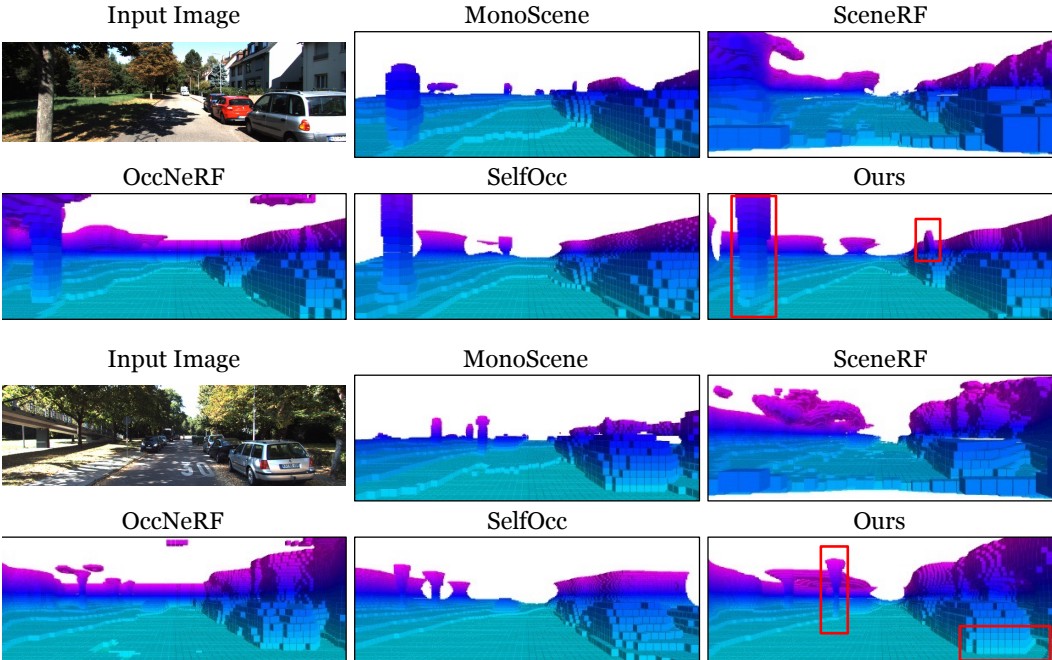

Figure 8: **Qualitative comparison for 3D occupancy prediction on the SemanticKITTI [35] validation set.** The red bounding box shows the most noticeable part.

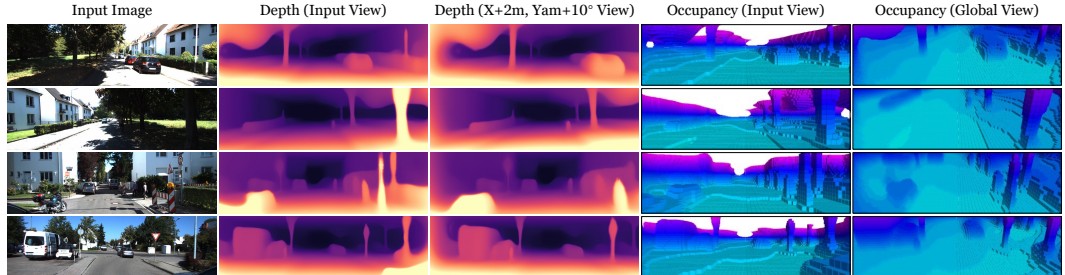

Figure 9: **Visualizations of depth estimation, novel view depth synthesis, 3D occupancy prediction on the SemanticKITTI [35] validation set.** (X+2m) means moving +2 meters along the x-axis of the LiDAR coordinate, and (Yaw+10°) means turning left for 10°.

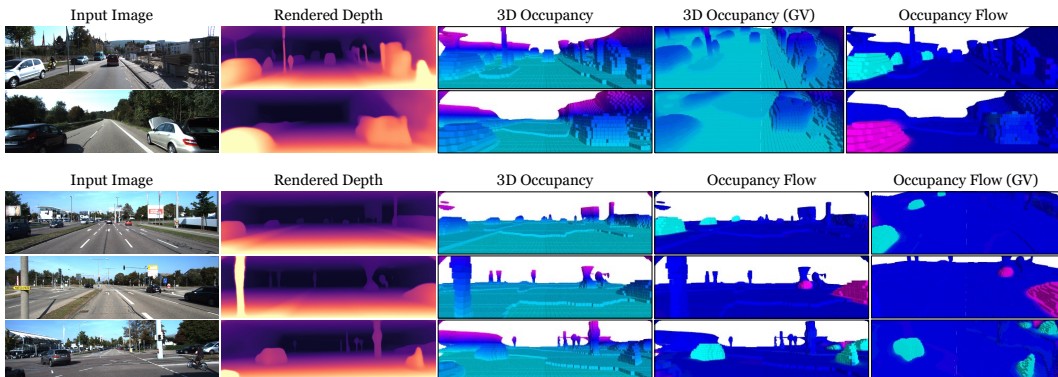

Figure 10: **Visualizations of depth estimation, 3D occupancy, and occupancy flow prediction on the KITTI-MOT [37] validation set.** GV indicates the global view.

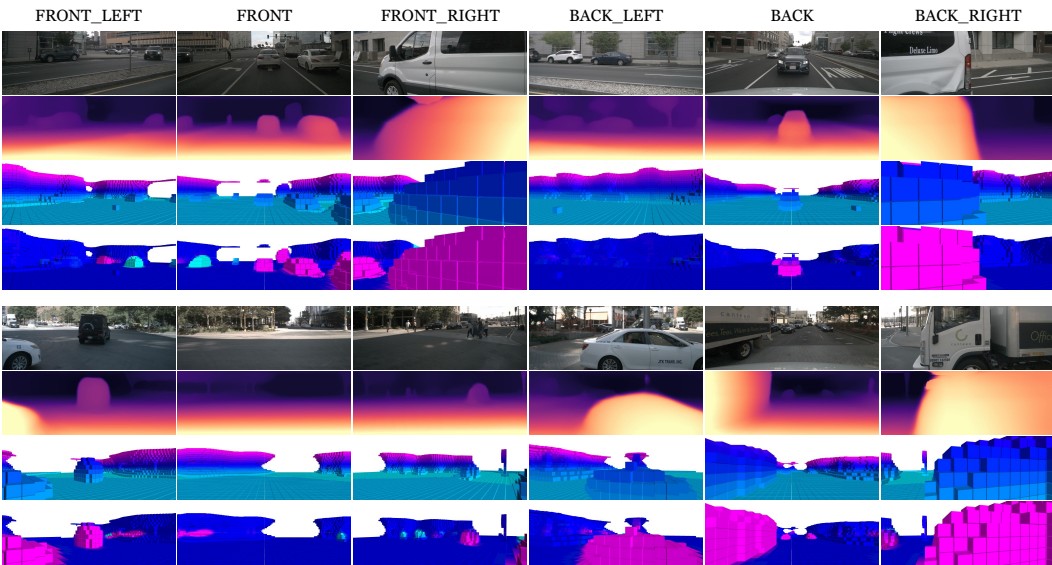

Figure 11: **Visualizations of depth estimation, 3D occupancy, and occupancy flow prediction on the nuScenes [39] validation set.** We show the six input surrounding images in the first row and the estimated depth from the corresponding views in the second row. The third and fourth rows demonstrate the predicted 3D occupancy and occupancy flow results separately.

