# OpenReview forum: "Let Occ Flow: Self-Supervised 3D Occupancy Flow Prediction"
_robot-learning.org/CoRL/2024/Conference — CoRL 2024_

### Official Review · Reviewer_PYze · 2024-07-19
**Self- and Weakly- Supervised 3D Occupancy Flow**

**Originality:** 3
**Technical Quality:** 4
**Clarity Of Presentation:** 3
**Potential Impact:** 3
**Recommendation:** 3
**Confidence:** 3

**Review:**

Strengths:
Scene-flow is a relevant task applicable in geometric scene understanding for autonomous navigation. Learning this in a self-supervised manner enables harnessing of larger and more diverse datasets without the need for expensive annotation.
The paper is generally well written and clear. The model architecture is explained clearly with diagrams and equations, and each step is motivated. The authors compare their method against other strong and recent methods in the field and outperform in most cases. The included ablation studies are useful to analyze the contribution of each of these innovations.

Weaknesses:
The model is not self-supervised in the sense it only uses raw sensor data to make predictions.
•	Weak annotation from pretrained optical flow models is utilized as a ground truth.
•	Semantic segmentation (Text + Grounded SAM) to identify moving objects for masking.
•	LiDAR is used whenever it is available in the ground truth data (this would be self-supervised in the flow sense, but not for occupancy/depth estimation).
Table 3’s column header (Supervision 3D/2D) could be somewhat misleading when the flow component of the task is not supervised, but the occupancy/depth is. Including a checkmark column to indicate usage of LiDAR and merging with Table 2 could be done (with a change from KITTI-MOT to nuScenes).
Since the main occupancy flow comparison between other methods uses nuScenes, the ablations should also use this.
Limitations should be included in the main paper rather than the supplementary.
This method isn’t shown on a robotics/autonomous vehicle system, hence, direct applicability of this computer vision algorithm on robotics hasn’t been demonstrated.

**Quality Of The Limitations Section:**

3

**Questions For Rebuttal:**

How does the number of adjacent frames used for the temporal fusion module trade off accuracy and runtime?
What is the proportional split in the two-stage optimization method between the first and second stage? How do you determine when to switch between them?
What impact does the 3d refine module have compared to interpolation for upsampling?
Could the authors clarify whether the A100 used for training were the 40GB or 80GB variants in Section B3.
What is the runtime inference cost of this system, would it be achievable to run on the typical compute budget of an edge robot or autonomous vehicle in real-time?

**Robotics Focus:**

2

**Summary Of Paper:**

This paper proposes a novel self-supervised 3d occupancy and flow prediction architecture. The model creates a “Temporal-Fused Volume” by geometrically aligning previous frames, processing with a Backward-Forward Attention Module on BEV compressed representations of the feature volumes and recombining with the original aligned volume. This is refined with a 3d convolution and then rendered with an MLP as a signed-distance-function or flow volume. A combination of loss functions is used including photometric reconstruction, l1-flow (from pretrained model), scale-invariant-depth loss from lidar (when available). To improve stability, a two-stage optimization process is employed to first optimize the SDF field with a static flow assumption, before learning flow. Further attention is paid to the imbalance between static and dynamic regions with “Dynamic Disentanglement” to apply a weighting factor to the loss in dynamic areas. Dynamic areas are found when geometric inconsistencies are found and combined with a pretrained segmentation model that has identified a dynamic object. The paper demonstrates SOTA performance in most fields and comparable settings, they also provide ablations showing the improvement each innovation contributes.

**Summary Of Recommendation:**

Learning 3D Occupancy Flow with self- or weak- supervision is an important area of study to reduce annotation intensity. The authors demonstrate SOTA on popular benchmark suites with their method. However, this method’s immediate applicability is not demonstrated on a live platform and there is key missing inference cost/latency information needed to gauge whether it can run on at the edge.

---

### Official Review · Reviewer_Dnni · 2024-07-19
**Strong paper on occupancy flow prediction for driving scenes, but with odd choices of evaluation settings**

**Originality:** 4
**Technical Quality:** 4
**Clarity Of Presentation:** 4
**Potential Impact:** 3
**Recommendation:** 3
**Confidence:** 4

**Review:**

Strengths:
- 3d occupancy flow prediction is an important under-explored task with various applications in robotics and autonomous driving
- The proposed architecture makes sense to me. It combines a number of recent advances (such as differentiable volumetric rendering, zero-shot segmentation) in a non-trivial manner.
- The paper is well structured and is mostly easy to read
- The method is evaluated in relevant settings, using multiple datasets, and according to the presented results the method outperforms SOTA techniques in all settings.

Weaknesses:
- My main concern is that the choice of evaluation settings and comparisons with prior works seems arbitrary, such that it makes it hard to draw a comprehensive conclusion. For example, 3d occupancy prediction results are only presented on KITTI data, whereas the two main baselines, OccNeRF and SelfOcc report results on NuScenes data. While flow prediction results are reported on nuScenes, there are no occupancy prediction results, so one cannot directly compare with previously reported results. Similarly, the choice of when to include lidar seems arbitrary. I am concerned that post-evaluation the authors have picked only the favorable settings where the proposed method worked well, and excluded settings where it did not. It would be important to either carefully justify these choices, or include results for all combinations of datasets/tasks/lidar-supervision. It is perfectly acceptable if the method does not outperform baselines in all settings.

Minor:
- It would be helpful to briefly explain metrics in the main text. For example, what does IuO@1 vs. IoI@2 refer to in Table 1?
- In line 152 i found the term "photometric loss" confusing. Is this a depth-only supervision or does it also include some visual (rgb) features?
- In Table 3, are the 3d annotations used for OccNet training come from an autolabelling tool or human annotations? If it's only autolabelling, i am not sure it's fair to classify it under "3d supervision" in contrast to the "2d supervision" used for "ours".
- How does the proposed method perform on nuScenes occupancy flow prediction without lidar?
- It is somewhat misleading to call the method self-supervised when it relies on a pre-trained segmentation model (SAM)
- Some recent works on self-supervised occupancy and flow prediction methods could be discussed, such as [1]

[1] Yang, Jiawei, et al. "Emernerf: Emergent spatial-temporal scene decomposition via self-supervision." arXiv preprint arXiv:2311.02077 (2023).

**Quality Of The Limitations Section:**

3

**Questions For Rebuttal:**

I would be open to increasing my scores if my concerns around experimental results above were adequately addressed.

----
After rebuttal I increased my score from "weak reject" to "weak accept"

**Robotics Focus:**

3

**Summary Of Paper:**

The paper proposes a novel 3d occupancy flow prediction network, leveraging zero-shot 2d segmentation from pre-trained foundation models. SOTA results are demonstrated on multiple datasets.

**Summary Of Recommendation:**

Potentially good paper, but i am concerned about experimental settings that makes it hard to compare with previously reported results.

---

### Official Review · Reviewer_hWju · 2024-07-20
**Though it proposes a model jointly predicts 3D occupancy and flow, no complete comparison to popular datasets and the state-of-the-art methods that provide reference to its performance was provided and not enough explanation or ablation is provided to dissect the contribution.**

**Originality:** 3
**Technical Quality:** 3
**Clarity Of Presentation:** 3
**Potential Impact:** 2
**Recommendation:** 3
**Confidence:** 3

**Review:**

### Strengths

- First to predict the occupancy and occupancy flow

### Weaknesses

- No semantics predictions were made which should be trivial. While most occupancy methods come with semantic predictions, the proposed method dropped the semantic segmentation part. One of the experiments was on SemanticKITTI but only worked on occupancy prediction. I believe semantic segmentation is as important as occupancy prediction in practice and dropping semantic part makes the method less practical.
- Datasets and baselines are not consistent. For instance, OccNerf has depth estimation and 3D occupancy prediction result on nuScenes dataset. However, the proposed method is only evaluated on SemanticKITTI while it uses nuScenes dataset result on flow estimation. While it uses a module from FB-OCC, it was not included in the evaluation. This does not provide the holistic view of the performance and makes it hard to compare its performance to the state-of-the-art methods. So it can be seen as an unfair comparison.
- Heavily rely on 2D flow supervision. While the loss terms for occupancy prediction follow the literature, the flow estimation relies on different losses from 2D flow supervision (except for the static flow supervision). This makes the proposed method look half-self-supervised.
- Hard to understand which architectural choice made a difference in performance and their novelty. It combines different modules from various papers. Is a different backbone contributed more than adopting TPV instead of BEV did? Is fusing BEV features instead of the volume features a novel idea or coming from another paper? As a reader, it was difficult to tell what contributed to the performance from the design.

**Quality Of The Limitations Section:**

3

**Questions For Rebuttal:**

In addition to the review, I have some questions:
- I wonder if there is a reason not applying BFAM to other methods in Table 3. It can be a good ablation with different modules and fair comparison.
- It was hard to understand what static flow supervision does. Does it provide 3D correspondences between adjacent frames? First, it does not sound like self-supervision; this is lifted supervision from externally computed 2D optical flow as a pseudo-ground-truth. Second, if the ego motion alignment is possible, I think a transformation can recover the static motion.
- Is it possible to drop the flow supervision? It provides good understanding of how good the true self-supervised model is or how effective 2D flow supervision is.
- Is there any reason that reprojection loss does not have the auto-encoding part? Adding a color prediction will enable auto-encoding.
- Have you tried to predict the static transformation to match the static part of the occupancy flow and the transformed occupancy? It may simplify the model as the flow prediction becomes residual flow prediction. Also, it can be used as an extra training signal to enforce the consistency between the occupancy and the flow.

**Robotics Focus:**

3

**Summary Of Paper:**

Proposed a model that jointly predicts 3D occupancy and occupancy flow from 2D optical flow supervision.

**Summary Of Recommendation:**

Recommended a weak reject due to its inconsistency in evaluation and lack of analysis to dissect the contribution. First, it is hard to determine which level of performance it has based on the provided evaluation.

---

### Decision · Program_Chairs · 2024-09-04

**Decision:**

Accept

**Comment:**

This paper proposes an approach for joint 3D occupancy and occupancy flow prediction. The reviewers acknowledge the importance of the problem being addressed. They find the paper well written and appreciate the combination of recent advances and good experiments. However, the reviewers have also raised important concerns regarding inconsistent baseline evaluations, concerns that it is not entirely self-supervised as claimed, novelty is not entirely clear as it combines many components from existing work, inconsistencies in the results on different datasets and ablations. Limitations should be also mentioned in the main paper. Each of the reviewers have listed several questions to be addressed in the rebuttal.
Post-rebuttal: Most of the reviewers' concerns have been sufficiently addressed. Please address the remaining suggestions in the final version.